# Experimental Evaluation of Pedestrian-Induced Multiaxial Gait Loads on Footbridges: Effects of the Structure-to-Human Interaction by Lateral Vibrating Platforms

**DOI:** 10.3390/s24082517

**Published:** 2024-04-14

**Authors:** Bryan Castillo, Johannio Marulanda, Peter Thomson

**Affiliations:** School of Civil and Geomatic Engineering, University of Valley, Cali 760001, Colombia; johannio.marulanda@correounivalle.edu.co (J.M.); peter.thomson@correounivalle.edu.co (P.T.)

**Keywords:** human–structure interaction, anthropic loads, human gait, synchronization, structural vibrations

## Abstract

The introduction of resistant and lightweight materials in the construction industry has led to civil structures being vulnerable to excessive vibrations, particularly in footbridges exposed to human-induced gait loads. This interaction, known as Human–Structure Interaction (HSI), involves a complex interplay between structural vibrations and gait loads. Despite extensive research on HSI, the simultaneous effects of lateral structural vibrations with fundamental frequencies close to human gait frequency (around 1.0 Hz) and wide amplitudes (over 30.0 mm) remain inadequately understood, posing a contemporary structural challenge highlighted by incidents in iconic bridges like the Millennium Bridge in London, Solferino Bridge in Paris, and Premier Bridge in Cali, Colombia. This paper focuses on the experimental exploration of Structure-to-Human Interaction (S2HI) effects using the Human–Structure Interaction Multi-Axial Test Framework (HSI-MTF). The framework enables the simultaneous measurement of vertical and lateral loads induced by human gait on surfaces with diverse frequency ranges and wide-amplitude lateral harmonic motions. The study involved seven test subjects, evaluating gait loads on rigid and harmonic lateral surfaces with displacements ranging from 5.0 to 50.0 mm and frequency content from 0.70 to 1.30 Hz. A low-cost vision-based motion capture system with smartphones analyzed the support (*T_su_*) and swing (*T_sw_*) periods of human gait. Results indicated substantial differences in *T_su_* and *T_sw_* on lateral harmonic protocols, reaching up to 96.53% and 58.15%, respectively, compared to rigid surfaces. Normalized lateral loads (*L_L_*) relative to the subject’s weight (*W*_0_) exhibited a linear growth proportional to lateral excitation frequency, with increased proportionality constants linked to higher vibration amplitudes. Linear regressions yielded an average R^2^ of 0.815. Regarding normalized vertical load (*L_V_*) with respect to *W*_0_, a consistent behavior was observed for amplitudes up to 30.0 mm, beyond which a linear increase, directly proportional to frequency, resulted in a 28.3% increment compared to rigid surfaces. Correlation analyses using Pearson linear coefficients determined relationships between structural surface vibration and pedestrian lateral motion, providing valuable insights into Structure-to-Human Interaction dynamics.

## 1. Introduction

The limitations of serviceability due to vibrations in civil structures, reported on various occasions in structural engineering, are often related to the dynamic coupling between anthropic loads and the dynamic response of these structures. The dynamic relationships between pedestrians and structures have been extensively documented in the literature for civil structures, such as stairs [1,2], grandstands [3,4], slabs [5,6], and footbridges [7,8,9,10,11,12,13,14,15,16]. Footbridges have become one of the most affected structures in terms of vibration serviceability limit state owing to their natural condition of spanning large distances and the progressive incorporation of lighter and more resistant materials in the construction industry, which generates relatively low masses and damping and increases technical and aesthetic demands [17,18,19]. Consequently, these conditions generate structures that are increasingly slender and flexible, increasing their dynamic sensitivity to excessive vibrations owing to the action of loads induced by human activity [20,21]. The events that occurred on the Millennium Bridge in London during its inauguration support the previously mentioned idea, as excessive vibrations occurred in this structure due to the lateral synchronization of forces induced by pedestrians crossing the bridge and the dynamic response of the structure in this direction. This coupling resulted in accelerations close to 0.25 g with displacements of up to 7.0 cm [22,23].

Dallard et al. [13] initially described the dynamic effects in the Millennium Bridge, and the review conducted by Zivanovic et al. [17] positioned the effects of dynamic Human–Structure Interaction (HSI) as one of the topics of increasing interest in the scientific community during the last two decades [24,25,26]. However, this event has not been configured as an isolated case because there are historical records of similar situations [27]. The oldest cases for which there is technical reference were the collapse of the Broughton Bridge in the UK due to the dynamic loads induced by the harmonic march of 60 infantry soldiers crossing the structure [28], the lateral vibration affectations on the Toda-Park Bridge in Japan [10], and the Solferino Bridge in France [29]. Additionally, more recent incidents have been reported in this type of structure caused by anthropic forces induced, such as the cases of The Wuhan Yangtze Bridge, China [30]; Pedro and Inês Bridge, Coimbra-Portugal [14,15]; Long Span Steel Bridge, Changi Airport, Singapore [11,12]; Le Ponte del Mare, Pescara, Italy [31]; Goodwill Bridge, Brisbane, Australia [32]; Eeklo Bridge, Belgium [8,9]; and the Folke Bernadotte Bridge, Sweden [33], among others [34,35,36,37,38].

These footbridges, which have been affected by HSI effects, promote the hypothesis of great urban regeneration on a global level, where isolated segments are interconnected with higher technical, aesthetic, architectural, and engineering requirements [39]. This innovative perspective led to a remarkable revolution in the design, construction, and management of these structures. However, their sensitivity to anthropic forces is noteworthy, considering that the dynamic behavior of footbridges is generally associated with frequencies lower than 10.0 Hz. The vertical and lateral frequency content characteristics of these structures, typically between 1.6 and 2.4 Hz, establish a focus of sensitivity to the coupling between their structural vibrations and the pedestrian gait dynamics, which is classified as the most recurrent source of excitation in the footbridges, with a typical frequency content between 0.8 and 1.2 Hz [40]. This implies a latent risk that footbridges and pedestrian gaits may be affected by a synchronization phenomenon.

The coupling effects of anthropic loads with the vibrations of pedestrian bridges are due to the adaptive [41], random [42,43], and nonstationary conditions of the human gait [17,44]. HSI effects in flexible surfaces are configured as an advanced case of synchronization and are composed of three main important aspects: changes in the dynamic properties of the structure due to pedestrians, Human-to-Structure Interaction (H2SI) [45,46,47]; interactions between pedestrians in crowds, Human-to-Human Interaction (HHI) [48,49,50]; and changes in the pedestrian gait due to the structure’s vibration, Structure-to-Human Interaction (S2HI) [18,19,43,51,52]. Although many researchers have concluded that studies and interest in these topics have increased considerably in recent decades, aspects related to the behavior of the loads induced by the human gait in response to structural vibrations have not been fully detailed or understood [9,53]; for example, the effects of structural vibrations on pedestrian-induced gait loads in vertical and lateral directions simultaneously.

Human-induced gait loads produce a total ground reaction force in each direction composed of two principal contributions: passive and active components of the HSI effects. The passive component is related to the dynamic human response to the vibration of the footbridge; in contrast, the active component is associated with the active motion of the pedestrian on the structure [42,54,55,56,57]. The structural vibrations control the dynamic coupling with human-induced loads in a vertical direction. For lateral vibrations, additional effects have been observed, including a synchronization phenomenon of pedestrians with the vibration of the structure and interaction between pedestrians in the case of densely occupied structures [57,58].

In an effort to push the boundaries of research in HSI, this paper presents a comprehensive study on the assessment of experimental multiaxial S2HI effects. Although researchers such as Nakamura et al. [59], Pizzimenti and Ricciardelli [60], Ingólfsson et al. [61], Bocian et al. [57,62,63], and, more recently, Zhang et al. [64], have made significant strides in this field of study, developing experimental campaigns to evaluate the human gait on vibrating lateral surfaces using instrumented treadmills or force plates, none of these studies have tackled the experimental evaluation of simultaneous vertical and lateral human loads as was performed in this investigation. This paper is structured as follows. Section 2 describes the experimental methods used in this study; five subsections were set up: general description, anthropometric information of the STs, experimental setup, data processing, and verification of the experimental setup. In Section 3, the results of the experimental gait tests on rigid and vibrating surfaces are presented and discussed. Finally, Section 4 presents the conclusions of this investigation.

## 2. Experimental Methods

### 2.1. Investigation General Design

This study experimentally assessed the spatiotemporal and kinetic behavior of human gait using an experimental test framework called the Human–Structure Interaction Multiaxial Test Framework (HSI-MTF) and a low-cost marker-based motion capture (MOCAP) system with smartphone devices. From prior data on the study population [65], seven test subjects (TS) were selected for inclusion in this research. This number was determined considering practical limits while ensuring that the chosen subjects reliably represented the variability of the population. Initially, STs were equipped with passive retroreflective markers. Then, the STs had an adaptive period of walking at a regular pace for 2.0 min with constant lateral vibration. Later, they executed gait protocols of 30.0 s each with different frequency and displacement for the lateral movement of the platform, while body kinematics, videos, and kinetic data were collected. The data collected during the tests were used to assess the vertical and lateral behavioral kinetics of human gait on surfaces with harmonic lateral motion in comparison with its behavior on a rigid surface.

### 2.2. Subjects of the Test

Seven STs (7 males, 23.0 ± 3.4 years, 172.9 ± 4.6 cm, and 71.9 ± 5.2 kg) volunteered to participate in this investigation. Their anthropometric information is shown in Table 1, according to the scheme shown in Figure 1. All STs were free of injuries for at least 1.0 month, had a body mass index (BMI) of <25.5 (normal weight level), and were aged 20–30 years. All tests involving human test subjects were carried out in accordance with the Code of Ethics of the World Medical Association (Declaration of Helsinki) for experiments involving humans and the Universidad del Valle Ethics Committee.

### 2.3. Assembly Setup

An experimental test framework called the Human–Structure Interaction Multiaxial Test Framework (HSI-MTF), positioned in the Laboratory of Seismic Engineering and Structural Dynamics (LINSE) at the Universidad del Valle, Cali, Colombia, was developed to evaluate the kinetic effects on human gait produced by surfaces with lateral harmonic motion. This framework allowed the generation of three main conditions: (i) the repetitiveness of human gait for the evaluation of its stochastic and random nature; (ii) the incidence of net lateral motion on test subjects; and (iii) the acquisition of the generated anthropic loads. One or two of these characteristics are present in the setups performed by Nakamura et al. [59], Ingólfsson et al. [61], Pizzimenti et al. [66], Carroll et al. [46], Bocian et al. [57], and more recently, Haowen et al. [67]. However, none of these studies performed a biaxial configuration for the lateral and vertical evaluation of the loads induced by the human gait under the effects of a surface with lateral motion, which is the condition addressed in this research. The HSI-MTF, schematically shown in Figure 2, consists of four main components, as listed in Table 2.

### 2.4. Data Collection Procedures

Experimental tests were conducted on the HSI-MTF for the STs, as shown in Figure 3. These tests involved acquiring the lateral and vertical forces induced by human gait on rigid and flexible surfaces with a sample time of 500 Hz. The human gait was performed at a constant velocity of 1.10 $m\cdot s^{-1}$ under lateral sinusoidal movements induced for the HSI-MTF of 5.0, 10.0, 20.0, 30.0, and 50.0 mm, with frequencies for each displacement ranging from 0.70 to 1.30 Hz in steps of 0.10 Hz, and a duration of 30.0 s for each frequency step. In addition, a low-cost MOCAP system was implemented using smartphone devices (Xiaomi Redmi 9T (Xiaomi, Beijing, China) with 48.0 Mpx and a sampling rate of 30.0 fps) and four passive retroreflective adhesive markers with a diameter of 12.0 mm that were attached at specific points of interest: the center of mass, ankle, heel, and metatarsus (Figure 3, middle). This technology was implemented with the aim of identifying the gait cycle times and displacements in relation to the behavior of lateral loads. Before the tests, each ST carried out 2.0 min gait pre-testing at their typical gait velocity (approx. 0.90 $m\cdot s^{-1}$). Seven static tests and 245 dynamic tests were performed.

### 2.5. Data Post-Processing

The multiaxial kinetics of human gait were obtained using the HSI-MTF, where the gait loads were measured in three dimensions. Lateral and vertical data were processed using custom algorithms in MATLAB (R2023a). The signals were re-sampled from the original 500 Hz to 20 Hz by applying a digital anti-alias in a glow pass filter. Points of interest were labeled in the sagittal plane (ankle, heel, and metatarsus) and coronal plane (center of mass—hips and HSI-MTF reference point), as shown in Figure 3. Initially, the vertical and lateral loads were acquired on rigid and flexible surfaces for each ST, as described in Section 2.4. Then, these load data were processed for a period of 30.0 s, approximately 25–30 complete gait cycles. Simultaneously, motion data of the feet and the center of mass were acquired for 30.0 s in order to calculate the average periods of the gait cycles and the behavior of the center of mass with respect to sinusoidal motions induced by the HSI-MTF.

### 2.6. HSI-MTF Lateral-Load Verification System

The lateral gait loads were measured during the tests, including the anthropic and inertial forces of the HSI-MTF. The lateral human gait load can be calculated using the equilibrium of forces according to D’Alembert’s principle [68].
(1)$$F_{Lateral\;Gait}=\sum_{i=1}^{4}F_{ML,i}-m_{HSI-MTF}\cdot a_{HSI-MTF}$$
where $F_{ML,i}$ denotes the measured load from the *i*th load cell and $m_{HSI-MTF}$ and $a_{HSI-MTF}$ are the mass and measured acceleration of the HSI-MTF, respectively. First, two sine waves of 7.50 mm at 1.0 Hz and 2.50 mm at 1.30 Hz were used as inputs in the system, and the lateral load and acceleration of the HIS-MTF were recorded without the action of any human gait. In this case, $F_{LateralGait}$ should be zero. The measured inertial force, which is the first right portion of Equation (1), coincided with the one calculated with $m_{HSI-MTF}\cdot a_{HSI-MTF}$, as shown in Figure 4 for the first proposed sine-wave signal, which indicates a zero for $F_{LateralGait}$, according to Equation (1). The goodness of fit (FIT, Equation (2)) and Pearson linear correlation coefficient (ρ, Equation (3)) were calculated to assess the accuracy of the HIS-MTF in acquiring the induced force in human gait and inertial force.
(2)FIT=1−FM−F^CFM−mean(FM)
(3)$$\rho =\frac{1}{N-1}\sum_{i=1}^{N}\left(\frac{F_{M,i}-\mu_{F_{M}}}{\sigma_{F_{M}}})(\frac{{\hat{F}}_{C,i}-\mu_{{\hat{F}}_{C}}}{\sigma_{{\hat{F}}_{C}}}\right)$$
where $F_{M}$ and ${\hat{F}}_{C}$ are the measured loads from the load cells and the calculated using the acceleration and mass of the HSI-MTF, respectively. *N* is the data number, and $\mu_{F_{M/C}}$ and $\sigma_{F_{M/C}}$ are the mean and standard deviation of $F_{M}$ or ${\hat{F}}_{C}$, respectively. For the proposed sine-wave signals, FITs of 0.852 and 0.844, and ρ of 0.989 and 0.988, respectively, were obtained for each signal. These evaluations demonstrate the accuracy and sensitivity of the proposed system.

## 3. Results and Discussion

### 3.1. Support and Swing Phase Periods of the Human Gait

This section focuses on the existing variability of the step frequency ($f_{s})$ in the human gait. $f_{s}$ is one of the primary parameters that provides information regarding the randomness of the loads induced during human gait. According to the research conducted by Nakamura et al. [59], Pizzimenti [66], and Ingólfsson et al. [61], the $f_{s}$ parameter is susceptible to variations due to changes occurring in human gait, particularly when STs are exposed to dynamic effects on surfaces with harmonic motion close to their fundamental gait frequency. Therefore, in this study, motion data of the left foot were acquired as described in Section 2.4 and Section 2.5, to obtain the periods of the support ($T_{su}$) and swing ($T_{sw}$) phases of the STs, both on a rigid surface and in the flexible surface protocols described in Section 2.4. The frame sequence shown in Figure 3 (bottom) corresponds to a test on the rigid surface of ST 1.
(4)$${PDF}_{T_{i}}=f\left(T_{i}|\mu_{T_{i}},\sigma_{T_{i}}\right)=\frac{1}{\sigma \sqrt{2\pi }}e^{\frac{-{\left(T_{i}-\mu_{T_{i}}\right)}^{2}}{2{\sigma_{T_{i}}}^{2}}},\;\mathrm{for}\;\forall \;T_{si}\in R$$

The frame sequences had a duration of 30.0 s. This time window allowed the acquisition of approximately 30 complete gait cycles, enabling a stochastic analysis of the $T_{su}$ and $T_{sw}$ gait periods of the STs. Using violin diagrams, in Figure 5, the normal probability density function (PDF) obtained for these periods on a rigid surface, according to Equation (4), using the average ($\mu_{T_{i}}$) and standard deviation ($\sigma_{T_{i}}$) of these swings (*i* = 1, $T_{1}$= $T_{sw}$) and support (*i* = 2, $T_{2}\;$= $T_{su}$) periods acquired at all STs (1→7) are presented (shown as a white circle), along with the first quantile of the standard deviation, in comparison to the obtained data.

The $T_{su}$ and $T_{sw}$ periods evaluated during human gait in the STs on the rigid surface were consistent with those presented in previous studies. Grieve et al. [69] conducted an analysis and parameterization of human gait on fifty test subjects, including both women and men. They determined that the swing time during gait was less than half of the apparent individual period, as is the case for over 70.0% of the STs studied. The parameterizations of swing time conducted on individuals with a gait speed of 1.10 m/s reported average periods of 0.43 s, deviating by 10.0% from the mean values reported in this study. In the late 1970s, Andriacchi et al. [70] conducted an experimental campaign with 70 test subjects ranging from 22 to 59 years of age, with a height of 173 ± 70 cm and weights of 757 ± 307 N. They determined support periods of 664 ms and swing periods of 521 ms for a gait speed of 1.10 m/s. According to the mean values of this research, there were overestimated swing periods by 8.0% and underestimated support periods by 22.0%. Similarly, Stenum et al. [71] reported an analysis of human gait in 32 individuals (22 males and 10 females) using a Motion Capture (MOCAP) system with pose estimation, indirectly determining support and swing periods of 0.75 ± 0.10 s and 0.46 ± 0.025 s, respectively. These values have differences of less than 12.0% compared with the $T_{su}$ and $T_{sw}$ values evaluated here using the MOCAP system with smartphone devices. Because of the importance of the support time during gait, as this is the period when loads are transmitted to the surfaces, Stacy et al. [72] used a wireless sensor system integrated into footwear to establish a correlation between gait periods ($1/f_{s}$) and average support periods, determining an average support value of 0.70 ± 0.026 s, which differs by under 18.0% from those reported in this study. Cao et al. [73] recently conducted a gait analysis on lightweight and large-span structures to evaluate the influence of gait loads on the development of HSI effect phenomena. For this purpose, tests were conducted on a force plate with 25 test subjects, and although the research focused on determining a parameterization of the induced loads relative to the $f_{s}$ of pedestrian, an average support period of 0.68 ± 0.10 s was determined, with a difference of 9.0% from the $T_{su}$ evaluated in this investigation.

Despite the low percentage differences compared with other investigations, as shown in Figure 5 (top: support time; bottom: swing time), the results of human gait on a rigid surface suggest significant intra- and inter-personal spatiotemporal variability (ST 3, 4, and 6; shown in Figure 5) between the participants, emphasizing the need for additional experimental assessment. This condition of relative variability within everyone becomes even more critical when comparing walking on a rigid surface with walking on surfaces with harmonic lateral movements because of noticeable changes in the support and swing phases generated by the coupling of the walking of the STs with the harmonic dynamics of the excitations induced by the HSI-MTF. Figure 6 and Figure 7 present the variations in the periods $T_{su}$ and $T_{sw}$, respectively, for displacements (5.0 to 50.0 mm) and lateral frequencies (0.7 to 1.3 Hz) induced in the gait of all evaluated STs, according to the proposal in Section 2.4. In these figures, a forest plot diagram is used, where white dots represent the mean values obtained for the indicated displacements and frequencies. In addition, the size of the blue boxes varied inversely proportional to the standard deviation of the respective test, and the horizontal red lines indicate the first calculated standard deviation. Although in the case of ST 1 the variations for the period $T_{su}$ are minimal compared to the average of all STs on a rigid surface, significant changes occur in the period $T_{sw}$ for displacements of 20.0 and 30.0 mm, indicating a brief implication of the coupling effects that alter gait dynamics. This is further supported by the remaining STs, which show decreases of up to 96.53% and 58.15% for the periods $T_{su}$ and $T_{sw}$, respectively, for some induced displacements and excitation frequencies compared with the overall average obtained on a rigid surface for each ST. These results align with those of studies on laterally instrumented treadmills reported by Nakamura et al. [59], Sun and Yuan [30], Pizzimenti et al. [66], Ingólfsson et al. [61], Ricciardelli et al. [74], Bocian et al. [63], and Haowen et al. [67]. Furthermore, they are consistent with investigations on some pedestrian bridges in a similar analytical context, such as those conducted by Butz et al. [75] and Rönnquist [76]. In both cases, an alteration in the $f_{s}$ of the STs was reported as the frequency and magnitude of the induced lateral displacement increase or an increase in anthropic loads. These reports indicate an alteration of the natural gait kinematics and, consequently, of $f_{s}$, consistent with the behavior presented in this assessment of the human gait in this study.

### 3.2. Assessment of Human Gait-Induced Loads

Human gait is described as a specific form of locomotion in which the weight of the human body is alternately transferred and supported by both lower limbs [77]. These displacement conditions generate dynamic loads known technically as Ground Reaction Forces (GRF), which, when interacting with civil structures with dynamic sensitivity conditions, such as pedestrian bridges, can induce dynamic coupling phenomena (Lock-In Effects) that increase the probability of structural vibrations, conditioning their serviceability. The study of these coupling phenomena has significantly increased in recent years, focusing especially on the nature and behavior of lateral and vertical loads induced by human gait in direct interaction with structures. In this study, the assessment of loads induced by human gait on vibrating flexible surfaces was divided into two main components: lateral and vertical loads.

#### 3.2.1. Lateral Load

In accordance with Section 2.5, a 30.0 s evaluation of lateral loading induced by walking on a rigid surface was conducted for the seven STs. The induced lateral load (*L_L_*) was normalized with respect to the weight (*W*_0_) of each ST, yielding an average normalized lateral load of 2.0% and a maximum of 3.7%. These values are depicted in Figure 8 (top) for all evaluated STs within a 5.0 s range. Furthermore, these *L_L_* loads were processed using a cross-power spectral density (CPSD, Equation (5)) as proposed in [78,79,80].
(5)$${CPSD}_{xy}\left(\omega \right)=\sum_{m=-\infty }^{\infty }R_{xy}\left(m\right){\cdot e}^{-j\omega m};\;R_{xy}=E\left\{x_{n}y_{n}^{*}\right\}\;\forall \left\{-\infty <n<\infty \right\}$$

In the implementation of CPSD, both $x_{n}$ and $y_{n}^{*}$ correspond to the *L_L_* load of the i-th ST. Additionally, E{*} denotes the expected value operator. Based on the aforementioned, the average fundamental lateral gait period was identified as $f_{s}$ = 0.73 ± 0.10 Hz, the second gait harmonic as $f_{2}$ = 1.45 ± 0.15 Hz, and the third gait harmonic as $f_{3}$ = 2.20 ± 0.15 Hz for all the STs, as illustrated in Figure 8 (middle). Conversely, in Figure 8 (bottom), the normalized *L_L_* load induced by ST 1 on the lateral harmonic surface is considered, featuring a frequency of 1.0 Hz and an amplitude of 20.0 mm. The objective was to showcase the sinusoidal behavior of *L_L_*, its phase in relation to the harmonic lateral movement, and the anisotropic behavior of its magnitude compared to that recorded on a rigid surface [46]. This aligns with the characteristics of S2HI effects elucidated by Butz et al. [75], Rönnquist et al. [76], and Heinemeyer et al. [81]. These findings provide a theoretical foundation for implementing an *L_L_* lateral load model based on a Fourier sine series with dynamic load factors (DLF), pedestrian weight (*W*_0_), and gait frequency (*f_m_*), as proposed in Equation (6).

This load model was employed in one of the initial pedestrian bridge design guidelines, the French SETRA [82], with a recommended constant DLF (*n* = 1) of 0.05 for a pedestrian with weight *W*_0_ = 700 N and gait frequency *f_m_* = 2.0 Hz.
(6)$$L_{L}\left(t\right)=DLF\cdot W_{0}\cdot \sum_{i=1}^{n}\sin(2\pi \cdot i\cdot \left(\frac{f_{m}}{2}\right)\cdot t)$$

Literature reviews conducted by Zivanovic et al. [17] and Racic et al. [45] presented two pioneering investigations concerning the lateral modeling of human gait based on DLFs. Initially, reference is made to the work by Schulze (later expanded upon by Bachmann and Ammann [67]), which reported the values of $\alpha_{1}$ = 0.039, $\alpha_{2}$ = 0.010, $\alpha_{3}$ = 0.043, $\alpha_{4}$ = 0.012, and $\alpha_{5}$ = 0.015 as lateral DLFs for human gait interacting with civil structures at a *f_m_* = 2.0 Hz. Similarly, Bachmann et al. [83,84] reported values of $\alpha_{1}$ = 0.012 and $\alpha_{3}$ = 0.015 for *f_m_* = 2.0 Hz. However, these values are conditioned by the acquisition on rigid surfaces or the effects of S2HI relative to the studied civil structure, as in [85]. Nakamura et al. [59] developed a device consisting of a shake table and a force plate and reported the following maximum DLFs for five test subjects: {0.08, 0.11, 0.12, 0.13}, {0.09, 0.11, 0.13, 0.17}, {0.10, 0.13, 0.16, 0.18}, and {0.11, 0.15, 0.18, 0.21}. These values were obtained in an experimental campaign combined with displacements of {10.0, 30.0, 50.0, and 70.0} mm and lateral oscillation frequencies of {0.75, 0.87, 1.0, and 1.25} Hz, respectively. However, these reported DLF values are associated with static gait, limiting their applicability in representing continuous gait in civil structures.

The utilization of instrumented treadmills marked a significant advancement for the continuous measurement of forces induced by human gait over time, unencumbered by restrictions on stride length owing to test area dimensions. Sun and Yuan [30] integrated a treadmill, a force plate, and a shake table, and seven STs were assessed using this framework at two walking speeds (0.83 and 1.00 $m\cdot s^{-1}$), subjected to combinations of lateral vibrations with frequencies of 0.65–1.2 Hz and amplitudes of 4.0–50.0 mm. The expression $DLF\left(U_{0}\right)=1.18\times U_{0}+0.05\forall U_{0}<0.05\;m$ was introduced in this study; however, this formulation solely depends on the lateral displacement $U_{0}$, disregarding the significant effects of frequency changes, even for the same displacement value. This phenomenon, as demonstrated in this study and depicted in Figure 9, was observed.

A similar study was conducted by Pizzimenti [66], who instrumented a treadmill capable of executing predetermined combinations of amplitudes and frequencies. Five STs were evaluated in this research under three amplitudes (15.0, 30.0, and 45.0 mm) and five lateral vibration frequencies (0.60–0.92 Hz). Based on [60], the existence of two DLFs, intrinsic to gait behavior and self-excited lateral movement dynamics, was concluded. However, these results were not assessed around the critical lateral frequency (≈1.0 Hz) outlined in the literature for pedestrian bridges subjected to HSI, thereby limiting their application under real service conditions. Later, Ingólfsson et al. [61] performed the most extensive experimental pedestrian campaign to induce lateral forces by employing the experimental setup of Pizzimenti. In this study, measurements were taken from 71 individuals for various combinations of vibration frequencies (0.33–1.07 Hz) and lateral amplitudes (4.5–48.0 mm). However, their findings were presented in terms of pedestrian load coefficients proportional to the velocity and acceleration, denoted as $C_{p}$ and $Q_{p}$, respectively. Similarly, Carroll et al. [46] and Bocian et al. [63] employed instrumented treadmills to scrutinize the behavior of loads laterally induced by human gait. The former focused on indirectly estimating lateral loads through visual marker data, whereas the latter examined differences in lateral loads when developed with and without a virtual reality environment. In recent years, Haowen et al. [67] conducted a multi-axial load analysis on five test subjects by employing a treadmill, shake table, and pressure insoles to evaluate lateral excitations with variable frequencies (1.2–1.7 Hz) and amplitudes (10.0 and 20.0 mm). Although their analysis was centered on stride rates exceeding 2.0 Hz (running conditions), rendering it incomparable to human walking conditions, this research establishes a precedent for multi-axial anthropic load assessment.

Considering the aforementioned aspects, the present study emerges as a contribution for experimental campaigns of simultaneous lateral and vertical gait loads acquisition (utilizing the HSI-MTF) and assessment of lateral and vertical human-induced DLFs. The focal point of this investigation is the variation of these factors due to their interdependence with the lateral displacement amplitudes and frequencies. As evidenced by the outcomes depicted in Figure 9, these results show a linear growth in DLF values proportional to the escalation in the lateral excitation frequency. Additionally, an augmentation of this proportionality constant was identified in relation to the increase in the amplitude of the lateral vibrations. Moreover, parametric expressions capturing the interdependence between the lateral excitation frequencies and DLFs were established for each lateral amplitude value, yielding an average R^2^ value of 0.815 for linear regressions of the form $\mathrm{DLF}=M*f_{n}+b$. In addition, lateral excitation amplitudes equal to or exceeding 50.0 mm induce alterations in pedestrian gait kinematics as individuals strive to maintain the center of mass stability during vibrations. This adjustment substantially heightens the laterally induced force from pedestrians (an 82.6% increase compared to lateral excitations with 40.0 mm amplitudes).

#### 3.2.2. Vertical Load

The total vertical load (*L_V_*) induced by the gait of the STs on a rigid surface under the influence of lateral vibrations was acquired and assessed in this study. The acquisition was conducted using the HSI-MTF, which constitutes a novel comprehensive development for the multiaxial acquisition of anthropic loads that has not been previously documented in the literature. As outlined in Section 2.5, a 30.0 s evaluation of the *L_V_* induced by gait on a rigid surface was performed for the seven STs. The *L_V_* load was normalized with respect to the weight *W*_0_ of each ST, resulting in an average normalized vertical load of 110.0% and a maximum of 123.0%. These values are presented in Figure 10 (top) for all STs within a 3.0 s range. Furthermore, these *L_V_* loads were processed using cross-power spectral density in accordance with Equation (5), as elaborated in Section 3.2.1. Then, the average fundamental vertical gait period was identified as *f_sv_* = 1.45 ± 0.25 Hz, the average second vertical harmonic as *f_v_*_2_ = 2.90 ± 0.30 Hz, and the average third vertical harmonic as *f_v_*_3_ = 4.35 ± 0.45 Hz for all STs, as illustrated in Figure 10 (middle). Additionally, a noticeably lower fundamental lateral gait frequency was discernible when compared to those presented in the vertical direction. This observation arises from the analysis of the total vertical force. Moreover, in Figure 10 (bottom), the normalized *L_V_* load induced by ST 1 on the lateral harmonic surface is considered, featuring a frequency of 1.0 Hz and an amplitude of 20.0 mm. The objective was to demonstrate the typical behavior of a sinusoidal *L_V_* series and its anisotropic behavior, both in phase, with respect to lateral vibrations, and in magnitude.

During lateral vibrations, encompassing combinations of amplitudes and frequencies, the maximum normalized *L_V_* loads induced for the STs were identified over 30.0 s intervals, as detailed in Section 2.4. These assessments revealed that for vibration amplitudes below 20.0 mm across the entire evaluated frequency range, the normalized *L_V_* loads remained relatively constant (Figure 11), akin to those recorded on a rigid surface. However, for lateral amplitudes equal to or exceeding 30.0 mm, a linear increasing trend in normalized *L_V_* proportional to the rise in lateral excitation frequency was observed. Additionally, the augmentation of this proportionality constant with respect to the increase in the lateral vibration amplitude was determined beyond this critical amplitude value. Furthermore, parametric expressions capturing the interdependence between lateral excitation frequencies and the maximum normalized *L_V_* values for lateral amplitudes of 30.0 mm and 50.0 mm are established; these expressions demonstrate maximum increases of 26.5% and 28.3%, respectively, compared to a rigid surface.

### 3.3. Correlation between CM and Treadmill Movements

The impact of flexible surface dynamics on the behavior and development of anthropic activities has been a topic of widespread interest in structural engineering and one of the most promising research avenues in the coming years. The work undertaken by Fujino et al. [10] set a precedent in analyzing the dynamic coupling phenomena between pedestrian bridges and pedestrians traversing them. Through image processing, they determined that approximately 20.0% of pedestrians exhibited interaction effects with the lateral vibrations of the Toda Bridge in Japan, which exhibited a frequency behavior close to that of human gait in that direction. In subsequent years, image processing from motion capture systems employed in the study of interaction phenomena has focused on indirectly determining loads induced by anthropic activities [10,46,86,87]. Similarly, motion capture systems have been utilized in various studies with instrumented treadmills [63,64,66]. However, the objectives of these investigations were primarily focused on analyzing the kinematic behavior of the ST.

In this context, the present research determined existing correlation levels between the induced lateral vibrations (HSI-MTF reference control point) and the estimated center of mass (CM) displacements for each ST. The tracking points are shown in Figure 3 (top, right). The marker-based motion capture system used and implemented is rated in Section 2.4 and Section 2.5. The correlation values were calculated for the seven STs based on Equation (3), considering different combinations of HSI-MTF lateral vibration frequencies and amplitudes, as illustrated in Figure 12. Within these correlation surfaces, a distinct and specific behavior is established for each ST, which is an important detail to note given that even under the same lateral vibration conditions, STs experience varying S2HI effects. Maximum correlation levels of ρ=12.0% were obtained for ST 1 during lateral vibrations with an amplitude of 10.0 mm and a frequency of 0.8 Hz. By contrast, ST 5 exhibited a maximum correlation of ρ=60.0% for vibrations with amplitudes of 30.0 mm and frequencies exceeding 1.0 Hz. These experimental findings establish a foundation for understanding the kinematics of each pedestrian under the effects of S2HI.

## 4. Conclusions

A Human–Structure Interaction Multiaxial Test Framework (HSI-MTF) was developed and used to investigate the effects of Structure-to-Human Interaction (S2HI). The framework consisted of an instrumented treadmill placed on a shake table. The HSI-MTF allowed for the measurement of the pedestrian-induced gait loads in vertical and lateral directions simultaneously while the walking surface was subject to lateral sinusoidal motion. During the experimental program, the subjects walked at a constant pace and were exposed to a combination of lateral vibrations with varying amplitudes and frequencies. The primary conclusions are summarized as follows:

The support ($T_{su})$ and swing ($T_{sw}$) times of human gait exhibited differences of up to 96.53% and 58.15%, respectively, when comparing them between pedestrians on lateral harmonic moving surfaces to those on rigid surfaces. These outcomes suggest significant alterations in gait kinematics owing to lateral vibration effects, resulting in decreased support times on the dynamic surface to maintain walking stability. Furthermore, correlation coefficients between the center of mass of the STs and a reference control point in the HSI-MTF were computed, revealing that S2HI effects are intrinsic to each pedestrian, even under identical lateral vibration conditions.

The maximum lateral loads (*L_L_*) normalized to the weight *W*_0_ of the STs exhibited a linear growth proportional to the increase in the lateral vibration frequency. Additionally, an increase in this proportionality constant was identified in relation to the elevation of vibration amplitudes. Furthermore, parametric expressions capturing the interdependence between the lateral excitation frequencies and DLFs are established for each lateral amplitude value. These expressions yielded an average R^2^ value of 0.815 for linear regressions of the form $DLF=Mf_{n}+b$. In addition, the maximum vertical loads (*L_V_*) normalized to the weight *W*_0_ of the STs remained constant in relation to those recorded on a rigid surface for lateral vibration amplitudes equal to or less than 30.0 mm. However, beyond this value, a linear growth is observed, which is proportional to the increase in the lateral vibration frequency. Furthermore, an increase in the proportionality constant was identified with respect to the elevation of lateral vibration amplitudes.

## Figures and Tables

**Figure 1 sensors-24-02517-f001:**
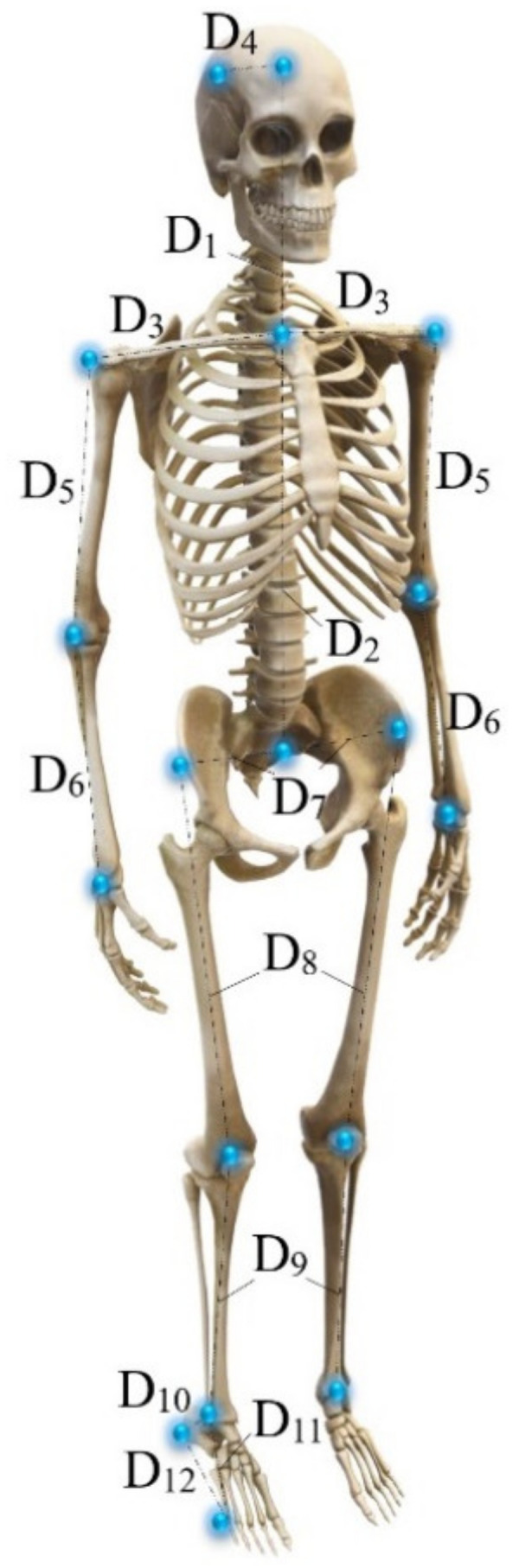
Schematic anthropometric information.

**Figure 2 sensors-24-02517-f002:**
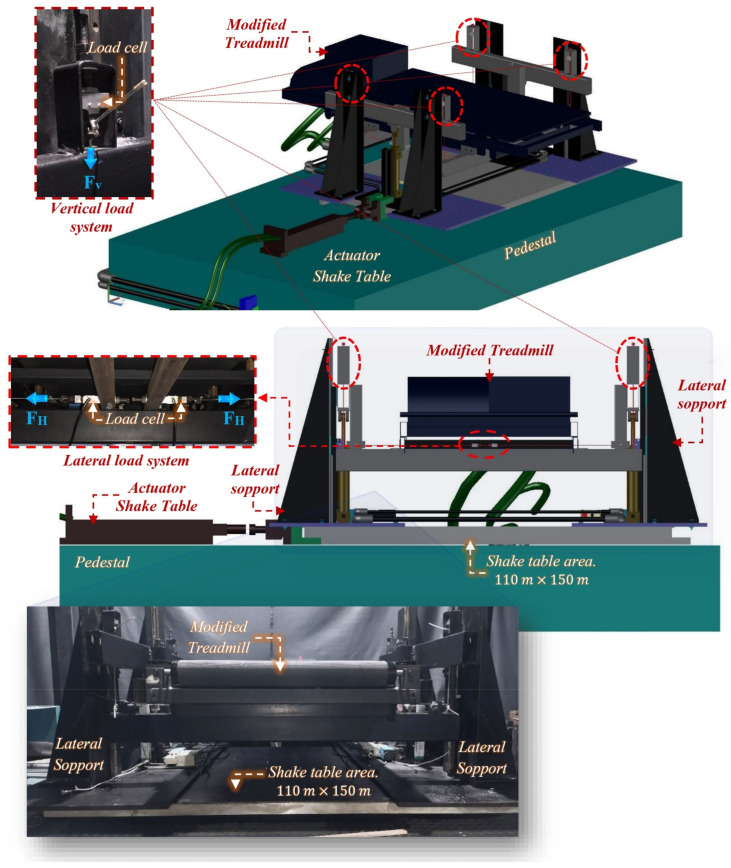
Human–Structure Interaction Multiaxial Test Framework: isometric view (**top**), frontal view (**middle**), real frontal view (**bottom**).

**Figure 3 sensors-24-02517-f003:**
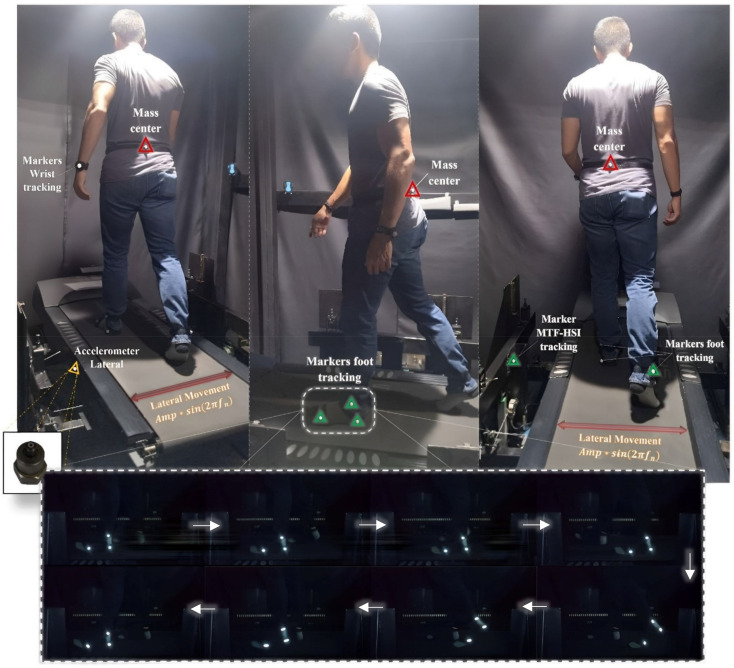
Experimental setup of the human gait test for ST No. 1. Isometric view (**top-left**), lateral view (**top-middle**), and top view (**top-right**). Additionally, a gait cycle (**bottom**).

**Figure 4 sensors-24-02517-f004:**
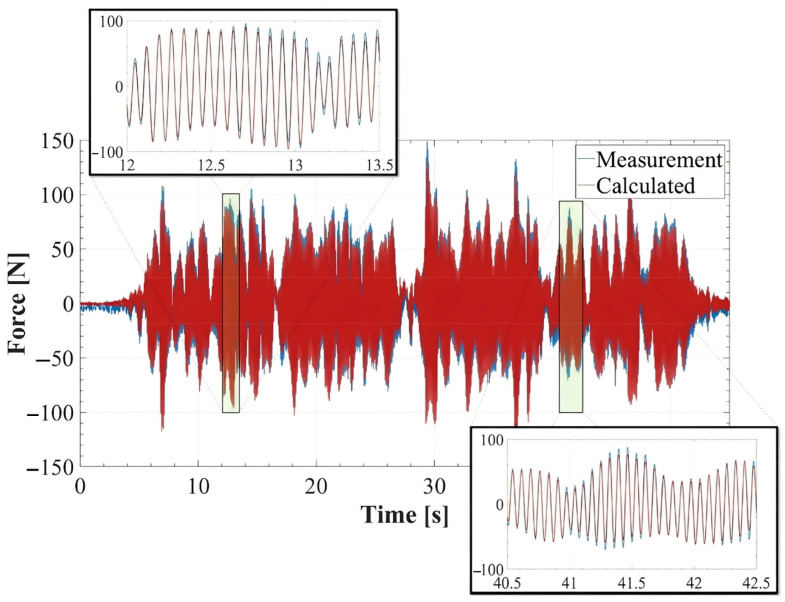
Inertial force calculated and measured by HSI-MFT.

**Figure 5 sensors-24-02517-f005:**
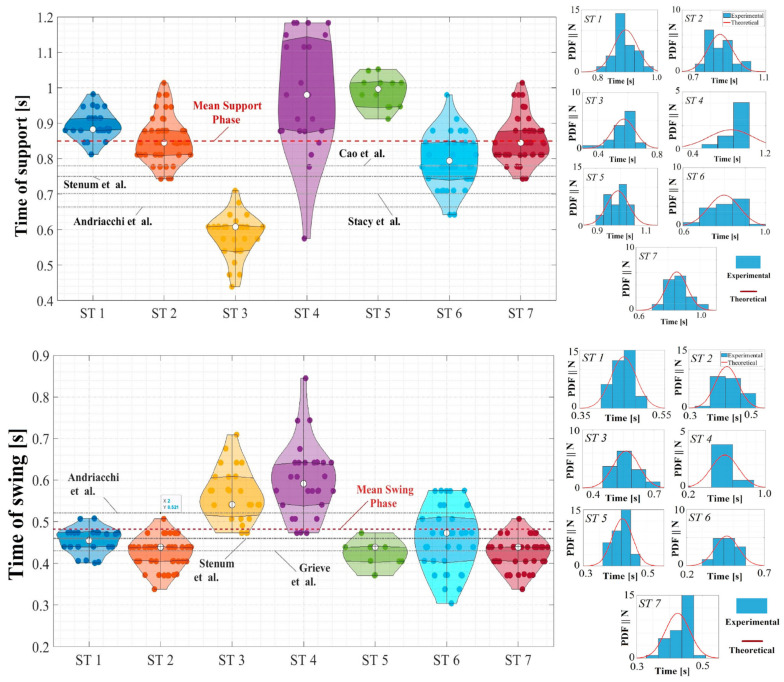
Statistical analysis of human gait support and swing periods; violin plot of the support phase duration and its probability density function for each subject test (**top**); and violin plot of the swing phase duration and its probability density function for each subject test (**bottom**); [51]—Grieve et al.; [52]—Andriacchi et al.; [53]—Stenum et al.; [54]—Stacy et al.; [55]—Cao et al.

**Figure 6 sensors-24-02517-f006:**
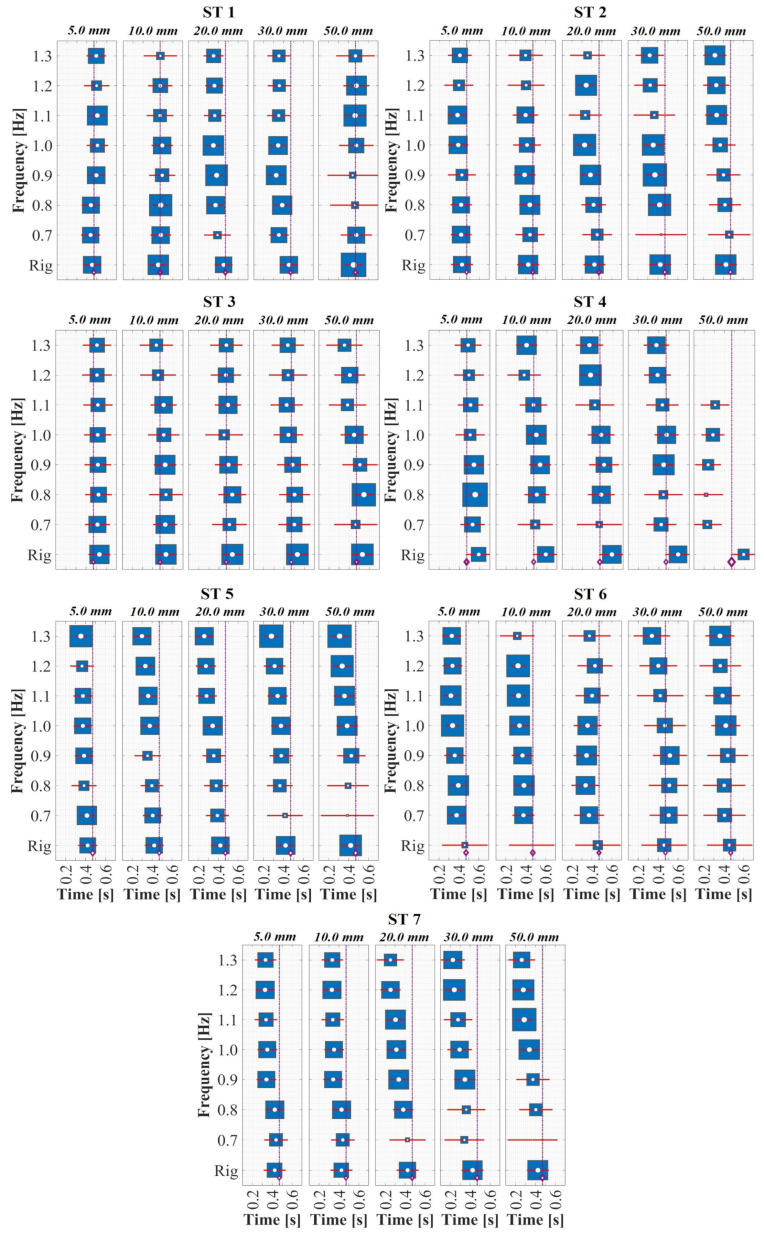
Support period ($T_{su}$) of the subject tests on multiplex harmonic surfaces.

**Figure 7 sensors-24-02517-f007:**
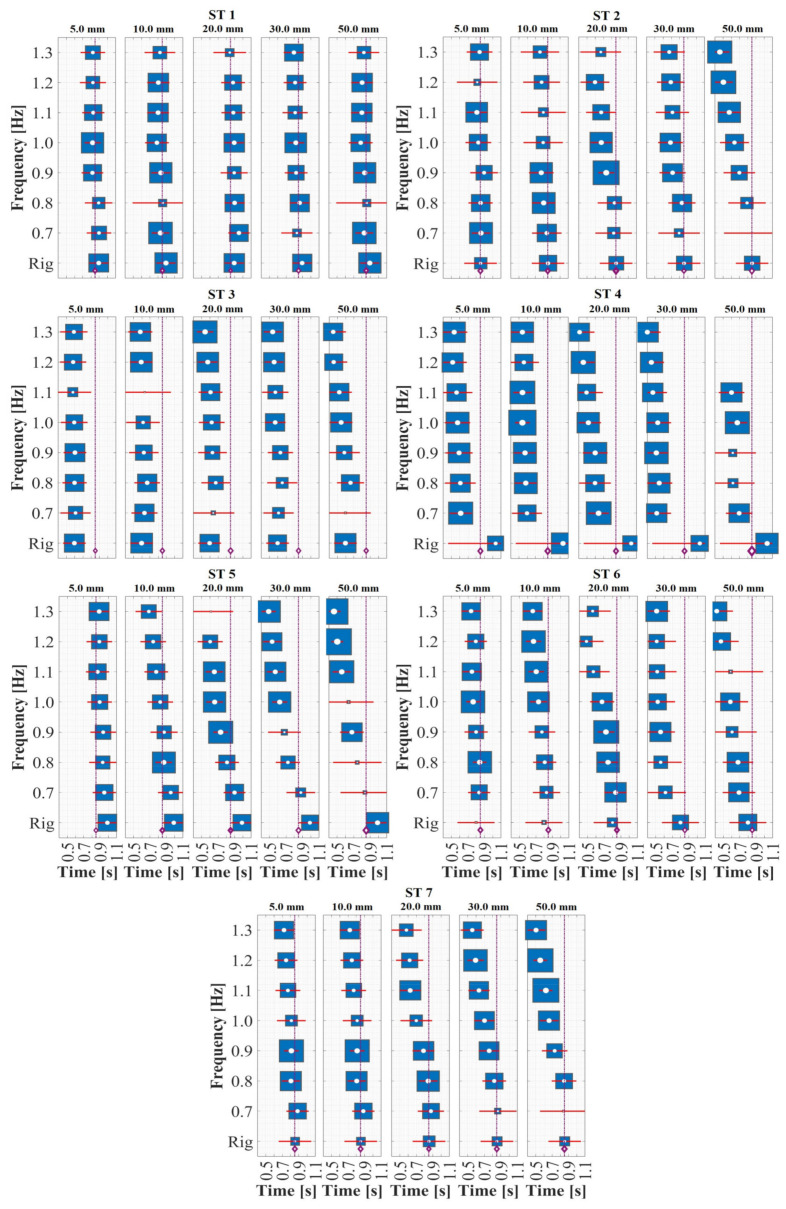
Swing period ($T_{sw}$) of the subject tests on multiplex harmonic surfaces.

**Figure 8 sensors-24-02517-f008:**
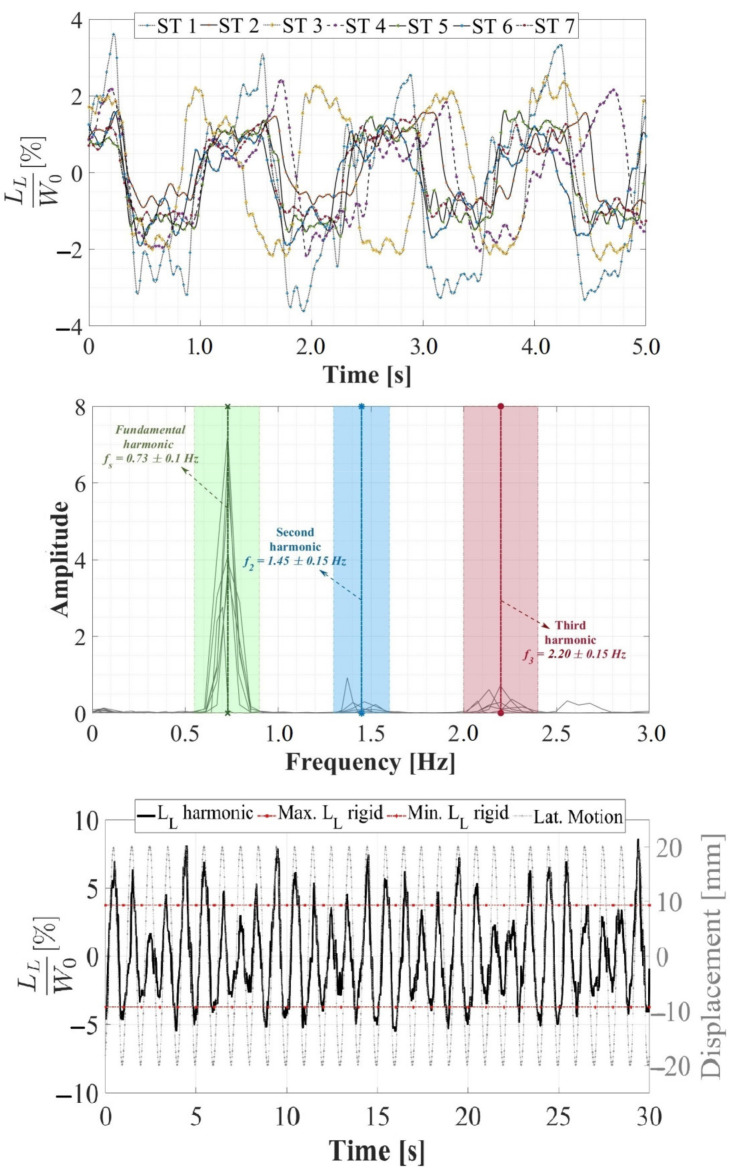
Lateral load induced by human gait on rigid and harmonic surfaces. Lateral loads (*L_L_*) normalized with respect to weight *W*_0_ induced by STs on rigid surfaces (**top**), frequency content of lateral loads induced by STs on rigid surfaces (**middle**), and lateral loads induced by ST 1 on harmonic surfaces with a frequency of 1.0 Hz and a displacement of 30.0 mm (**bottom**).

**Figure 9 sensors-24-02517-f009:**
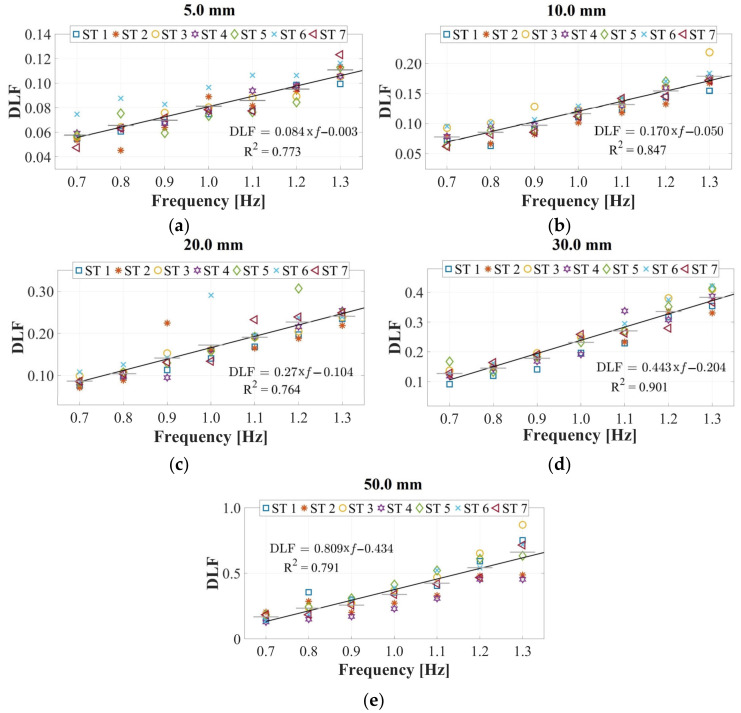
Experimental evaluation of pedestrian-induced lateral load on harmonic lateral surfaces with displacements of (**a**) 5.0 mm, (**b**) 10.0 mm, (**c**) 20.0 mm, (**d**) 30.0 mm, and (**e**) 50.0 mm.

**Figure 10 sensors-24-02517-f010:**
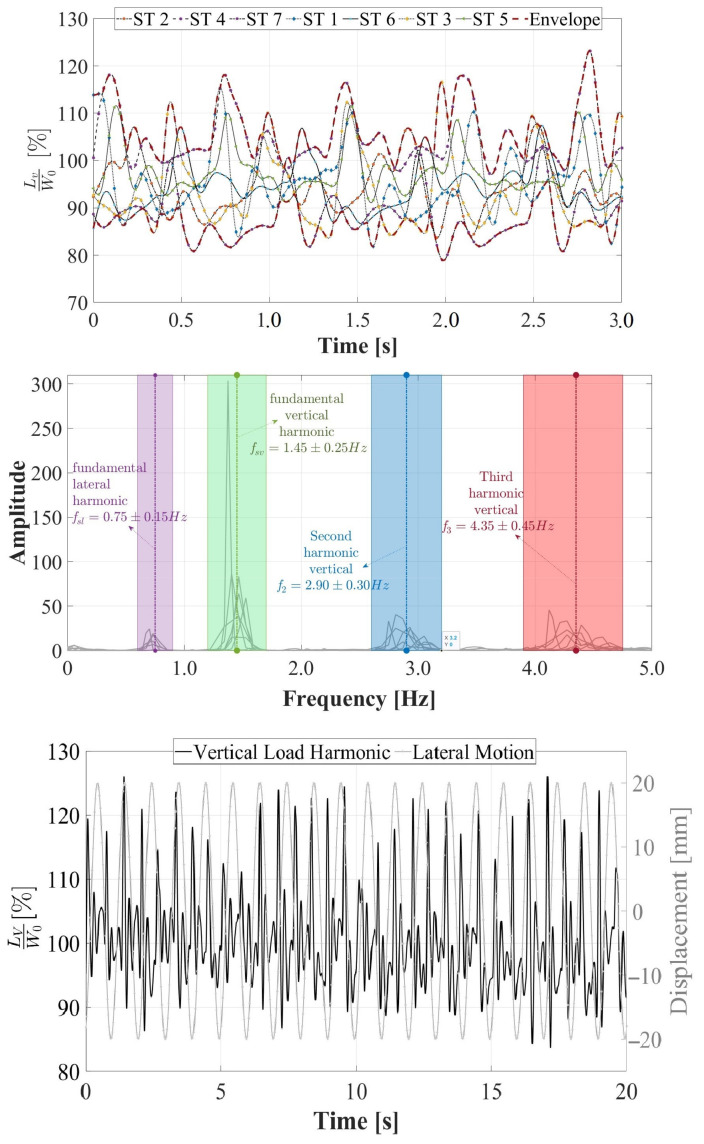
Vertical load induced by human gait on rigid and harmonic surfaces. Vertical loads normalized with respect to weight induced by STs on rigid surfaces (**top**), frequency content of vertical loads induced by STs on rigid surfaces (**middle**), and vertical loads induced by ST 1 on harmonic surfaces with a frequency of 1.0 Hz and a displacement of 30.0 mm (**bottom**).

**Figure 11 sensors-24-02517-f011:**
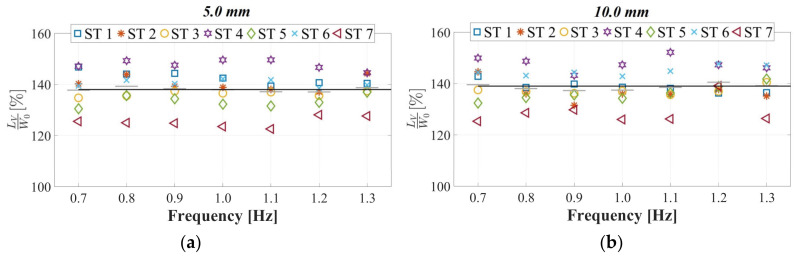
Experimental evaluation of pedestrian-induced vertical load on harmonic lateral surfaces with displacements of (**a**) 5.0 mm, (**b**) 10.0 mm, (**c**) 20.0 mm, (**d**) 30.0 mm, and (**e**) 50.0 mm.

**Figure 12 sensors-24-02517-f012:**
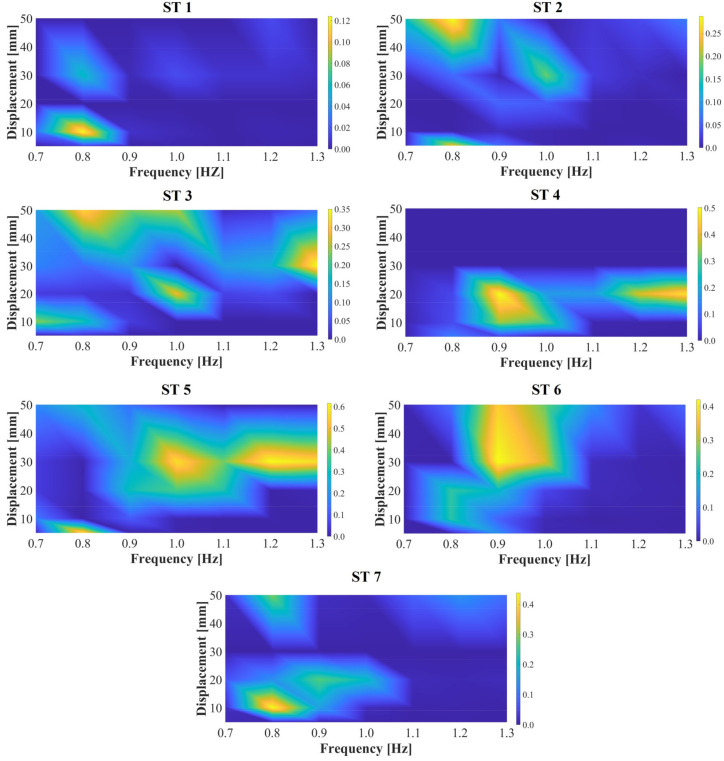
Correlation between the CM of the STs and the reference control point in the HSI-MTF.

**Table 1 sensors-24-02517-t001:** Anthropometric information.

		Test Subject						
		No. 1	No. 2	No. 3	No. 4	No. 5	No. 6	No. 7
Length (cm)	D1	27.2	24.7	23.6	27.0	28.0	31.0	35.0
	D2	47.5	47.5	47.2	42.0	57.0	55.0	54.0
	D3	22.3	19.8	21.7	20.0	22.0	20.5	40.0
	D4	18.7	16.7	15.8	8.0	9.0	6.0	13.0
	D5	31.4	30.6	30.2	28.0	30.0	23.0	27.0
	D6	26.7	27.4	26.7	24.0	23.0	25.0	27.0
	D7	17.7	19.2	19.8	19.0	22.0	17.0	33.0
	D8	47.5	43.0	44.2	48.0	48.0	48.0	45.0
	D9	46.7	46.5	45.8	43.0	43.0	42.0	41.0
	D10	9.8	10.4	10.1	9.0	9.0	11.0	12.0
	D11	17.5	15.5	15.2	15.0	15.0	15.0	14.0
	D12	19.4	18.6	17.9	17.0	18.0	18.0	18.0
Diameter (cm)	Head	57.0	58.0	54.0	56.0	56.0	53.0	58.0
	Shoulder	37.0	47.0	52.0	35.0	36.0	37.0	38.0
	Elbow	25.0	27.5	28.5	25.5	27.0	26.5	27.0
	Wrist	17.0	16.0	17.0	16.0	18.0	16.5	16.0
	Hip	90.0	85.5	88.5	91.0	104.0	91.0	92.0
	Knee	38.9	38.5	41.0	34.5	43.0	38.5	36.0
	Ankle	25.0	24.5	25.0	26.0	26.0	23.0	21.0
	Heel	32.0	23.0	23.0	30.5	33.0	34.0	30.0
	Metatarsus	24.0	24.0	54.0	22.0	23.5	27.5	58.0
General Informa.	Age	30	21	23	23	24	22	29
	Height (cm)	178	179	174	173	170	166	170
	Weight (kg)	70.0	74.0	76.0	70.0	80.0	69.0	64.0

**Table 2 sensors-24-02517-t002:** Main components HSI-MTF.

Item No.	Components Name	Description
1	Unidirectional shake table	The custom unidirectional shake table of the School of Civil and Geomatic Engineering, consists of an aluminum plate of 1.10 m × 1.50 m directed by low-friction linear rails and coupled to a Shore Western hydraulic actuator of 45.0 kN (Shore Western, Monrovia, CA, USA). This actuator has two DyVal servo-valves that are connected to a hydraulic potential of up to 977.9 cm^3^·s^−1^ at 21 MPa. In addition, it has an internal linear variable differential transducer (LVDT) that measures the position of the actuator rod. Recent work has allowed the identification of this system through frequency sweep tests from 0.10 to 20.0 Hz, with a duration of 250.0 s and constant amplitudes of 0.50, 1.0, 3.0, and 5.0 mm. A representation of the GHA plant of the seismic simulator was found to be a continuous system with a pole at the origin, a nominal gain of 14.0, and a time delay of 33.0 ms. In addition, this device was implemented with a robust H∞ displacement tracking controller conditioned by a Kalman filter.
2	Lateral support system	The lateral support system initially consisted of four custom ASTM A36 steel plates with a thickness of 1.27 cm, which were intended to increase the working area of component 1. Four lateral supports with a height of 74.0 cm were placed on top of the plates, which connected the guides of the vertical linear axes of the HSI-MTF. In addition, it supports a human gait loading acquisition system.
3	Treadmill	A SOLE brand treadmill model F65 was used (Sole Fitness, Salt Lake City, UT, USA), which has an effective action area of 72.0 cm × 150.0 cm and operating speeds ranging from 0.50 to 10.0 cm^3^·s^−1^. The forward motion is generated by a 120.0 V commercial motor controlled by the factory hardware. The treadmill was compacted and coupled with the HSI-MTF.
4	Load acquisition system	This system was developed to quantify the anthropic loads induced by human gait and consisted of a ∅1.59 mm steel wire with an ultimate load of 4.0 kN, attached to the lateral support system on the linear axes and to the bottom of the treadmill through four Omega DYLY-103 load cells (Omega Engineering, Norwalk, CT, USA) with an individual capacity of 200.0 kg and a sensitivity of 0.15 mV/kg. The vertical acquisition component was formed by a ∅3.18 mm steel wire with an ultimate load of 10.0 kN, attached to the lateral support system on the linear axes and the vertical suspension system through four Omega LC-703 load cells with a capacity of 90.70 kg each and a sensitivity of 0.20 mV/kg. Finally, the longitudinal acquisition component had the same steel wire attached to the most extreme parts of the vertical suspension system, using four Omega DYLY-102 load cells with a capacity of 50.0 kg each.

## Data Availability

The raw and time-normalized marker-based with Opti-Track and smartphone kinematic data, as well as the test video, are available from Mendeley Data at https://data.mendeley.com/datasets/6r53j3576r/1 (accessed on 18 January 2024).

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
