# Peer review of "Experimental Evaluation of Pedestrian-Induced Multiaxial Gait Loads on Footbridges: Effects of the Structure-to-Human Interaction by Lateral Vibrating Platforms"

_sensors, 2024, doi:10.3390/s24082517_

Round 1

Reviewer 1 Report

Comments and Suggestions for Authors

The paper treats the characterization of the lateral forces produced by humans, which are important to be known when the coupling with a pedestrian structure is considered.

In my opinion there are new interesting data presented in the paper.

However, the introductions lacks of a clear explanation of the phenomena involved: each walking human produces a total ground reaction force in each direction which is the sum of two contributions: a passive component which is related to the human response to the vibration of the pedestrian structure plus an active component linked to the active motion of the person [1–5]. For vertical vibrations of the pedestrian structure this is enough to explain the coupling with the forces produced by a walking person. Instead, for lateral vibrations, an additional effect arises, which is the synchronization of person motion with structure vibration [6]. When a dense crowd is occupying the structure, then, additional effects must be taken into consideration (i.e., synchronization of motion of different persons) [7].

1.         Van Nimmen K, Lombaert G, De Roeck G, Van den Broeck P. The impact of vertical human-structure interaction on the response of footbridges to pedestrian excitation. Journal of Sound and Vibration 2017; 402: 104–121. DOI: 10.1016/j.jsv.2017.05.017.

2.         Berardengo M, Drago L, Manzoni S, Vanali M. An approach to predict human – structure interaction in the case of staircases. Archive of Applied Mechanics 2019; 89(10): 2167–2191. DOI: 10.1007/s00419-019-01569-2.

3.         Venuti F, Racic V, Corbetta A. Modelling framework for dynamic interaction between multiple pedestrians and vertical vibrations of footbridges. Journal of Sound and Vibration 2016; 379: 245–263. DOI: 10.1016/j.jsv.2016.05.047.

4.         Lucà F, Berardengo M, Manzoni S, Vanali M, Drago L. Human-structure interaction: convolution-based estimation of human-induced vibrations using experimental data. Mechanical Systems and Signal Processing 2022; 167(April 2021): 108511. DOI: 10.1016/j.ymssp.2021.108511.

5.         Tubino F. Probabilistic assessment of the dynamic interaction between multiple pedestrians and vertical vibrations of footbridges. Journal of Sound and Vibration 2018; 417: 80–96. DOI: 10.1016/j.jsv.2017.11.057.

6.         Bocian M, Burn JF, Macdonald JHG, Brownjohn JMW. From phase drift to synchronisation ??? pedestrian stepping behaviour on laterally oscillating structures and consequences for dynamic stability. Journal of Sound and Vibration 2017; 392: 382–399. DOI: 10.1016/j.jsv.2016.12.022.

7.         Venuti F, Bruno L, Bellomo N. Crowd dynamics on a moving platform: Mathematical modelling and application to lively footbridges. Mathematical and Computer Modelling 2007; 45(3–4): 252–269. DOI: 10.1016/j.mcm.2006.04.007.

Furthermore, please check typos and missing references.

Comments on the Quality of English Language

--

Author Response

Dear reviewer,
We would like to express our gratitude for your time and valuable insight. We have modified the manuscript considering your recommendations and the suggestions from the other reviewer. We are confident that the revised version enhances the overall clarity and impact of the paper. Please find in the attached file a comprehensive list of answers to your observations.

Reviewer 2 Report

Comments and Suggestions for Authors

The research paper ‘experimental evaluation of pedestrian-induced multiaxial gait loads on footbridges: effects of the structure-to-human interaction by lateral vibrating platforms’,

 By Castillo et al,

 investigates the impact of Structure-to-Human Interaction (S2HI) in the context of footbridges exposed to human-induced gait loads. More specifically, the study addresses the challenges posed by the simultaneous effects of lateral structural vibrations, particularly those with frequencies close to human gait frequency and wide amplitudes. The research employs the Human-Structure Interaction Multi-Axial Test Framework (HSI-MTF) to experimentally explore S2HI effects.

 Overall, the paper is of interest for researchers in bridge monitoring, especially for human-structure interaction.

 Nevertheless, several issues, related to both the paper’s content and format, are enlisted here below and should be addressed in order to achieve full acceptance.

1.      The conceptual difference between HSI and S2HI is not totally clear to this Reviewer. If understood well, differently from classic HSI, which focuses on the effects of human (static or dynamic) loads on the infrastructure’s dynamic response, here the point of view is reversed, focusing instead on the effects of the bridge lateral oscillations on the pedestrian gait? The Introduction reports four references on this point (line 93), which are quite few. Is this topic so much underinvestigated?

2.      The Type of the Paper should be indicated not in brackets but replacing the sentence on the top left of the first page

3.      The title should not be all capital letters.

4.      If all Authors are from the same institute, there is no need to report them separately three different times. Please follow the guidelines for authors.

5.      The introduction does provide a detailed list of interesting recent incidents (lines 68-72), yet, the state-of-the-art review about HSI analytical models can be expanded. For instance, the very recent ‘Quantification of the human–structure interaction effect through full-scale dynamic testing: The Folke Bernadotte Bridge’ is not mentioned.

6.      Seven test subjects participated in the study. Is there any specific reason for this number of participants? Is this intended to guarantee statistical relevance to the outcomes and/or was it limited by mere practical issues? Generally, such tests require a very large cohort of individuals.

7.      Figure 2 reports the 3D graphical scheme of the machine. Is a picture of the whole setup available instead? In addition to the current figure. 

8.      Eq (1): ‘*’ generally indicates convolution in mathematical writing. If the intended meaning is a simple multiplication, use the dot symbol instead. The same issue in the Conclusions (line 492)

9.      Figure 5 is very well made and deserves praise. However, the caption should be expanded to better describe the graphical components of the statistical analysis

Comments on the Quality of English Language

The English of the paper is overall fine, nevertheless, a careful double-check is suggested. 

Author Response

(The authors gave the same response as above.)

Round 2

Reviewer 1 Report

Comments and Suggestions for Authors

The paper can be published as is.

Author Response

Dear reviewer,

We would like to express our gratitude for your time and dedication to improve this manuscript.

Best regards.

Reviewer 2 Report

Comments and Suggestions for Authors

This Reviewer is satisfied by the reply to the first round of comments. Content-wise, the manuscript is complete and solid. Only a few editorial (thus, minor) issues should be addressed by the Authors:

1.      Please be aware that some cross-reference links are not working (in the pdf: ‘Error! Reference source not found.’)

2.      Table 1, the measurement unit for height is indicated as (mm) but it clearly is (cm).

3.      As mentioned in the first round of comments, ‘*’ generally indicates convolution in mathematical writing. If the intended meaning is a simple multiplication, use the dot symbol instead. This issue was correctly solved by the Authors in Eq (1) and elsewhere in the text but not everywhere (see line 414)

4.      Line 68, the ‘and’ before Eekl Bridge should be removed.

5.      Related to the previous remarks, a double-check of the whole manuscript, including not only the main text but also the content and caption of Tables and Figures, is strongly advised.

Comments on the Quality of English Language

The English of the paper is fine

Author Response

(The authors gave the same response as above.)
